# Watershed-enhanced cell instance segmentation based on guidance of gradient

**Zhengwei Lu**
Department of Computer Science
Zhejiang University of Technology
Zhejiang, China 310014
2111912083@zjut.edu.cn

**Zehan Zhang**
Zhejiang University of Technology
Zhejiang, China
2112012119@zjut.edu.cn

## Abstract

Cell Segmentation in Multi-modality Microscopy Images NeurIPS 2022 competition aims to benchmark cell segmentation methods that could be applied to various microscopy images across multiple imaging platforms and tissue types. Due to the difficulty of obtaining labeled data, the competition team set this cell segmentation problem as a weakly supervised task and provided some labeled data and a large amount of unlabeled data. We constructed a network structure with CoaT as the encoder and designed a decoder for fusing different scale features, and conducted related experiments with full supervision and semi-supervision respectively. We followed cellpose's strategy and constructed a model to predict the central region of individual cells, and obtained the final instance segmentation results by the watershed algorithm. The method we proposed obtained an F1 score of 0.7724 on the tuning set.

## 1 Introduction

Many biological applications require the segmentation of cell bodies, membranes, and nuclei from microscopy images. Cellpose [1] generated topological maps through a process of simulated diffusion that used ground-truth masks. A neural network was then trained to predict the horizontal and vertical gradients of the topology map and a binary prediction mask to indicate whether a given pixel was inside or outside the ROI. On the test image, a vector field is also obtained by predicting the binary mask as well as the horizontal gradient and vertical gradient by the network, and the exact shape of individual cells is recovered by this vector field to obtain the final instance segmentation results.

There are some differences between the universal instance segmentation of natural image and cell instance segmentation. We summarize this task's key points and difficulties: (1) Universal Instance Segmentation always has multiple class instances in each image, but cell segmentation is different. Most of the time we study only a few or a single class of cells under the same imaging instrument. (2) The target cells sizes are relatively balanced over the whole dataset, but the cell size is stable in the same image during cell segmentation. (3) The density of image instances is different, and the task of dense segmentation of the same type of instances is performed during cell segmentation. (4) Boundary separation is more difficult. In cell segmentation, the area close to the cell membrane can hardly be distinguished by significant edge features. Shape information in a wide field of view is required, which requires a large receptive field.

We followed cellpose's strategy and constructed a model to predict the central regions of individual cells indirectly, and then to obtain the final instance segmentation results by the watershed algorithm

36th Conference on Neural Information Processing Systems (NeurIPS 2022).

to enhance concept of the "cell's center" compared with vanilla cellpose. For the architecture part, we used CoaT-Lite [2] as the encoder of the network and designed a decoder to fuse the outputs of different layers of the encoder. In addition, to obtain a model with better generalization capability, we use the mean-teacher approach to perform semi-supervised training using the provided unlabeled data.

The structure of this paper is as follows: Chapter 2 introduces the proposed method in three parts: preprocessing, model architecture, and post-processing. The datasets used and the details of the experiments are described in detail in Chapter 3. Chapter 4 presents the results of some of our experiments and a discussion about them. In the last chapter, we summarize the task and the competition process.

## 2 Method

In this chapter, we will detail the strategy and process of the adopted method in three parts: pre-processing, model architecture, and post-processing.

### 2.1 Preprocessing

To follow cellpose's instance segmentation strategy, we process the provided instance labels to obtain the corresponding gradient vector fields and binary classification masks. Fig 1 shows an example of pre-processing the instance labels.

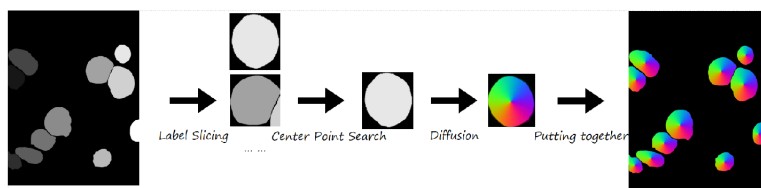

Figure 1: Instance of splitting labels to generate gradient fields.

### 2.2 Model Architecture

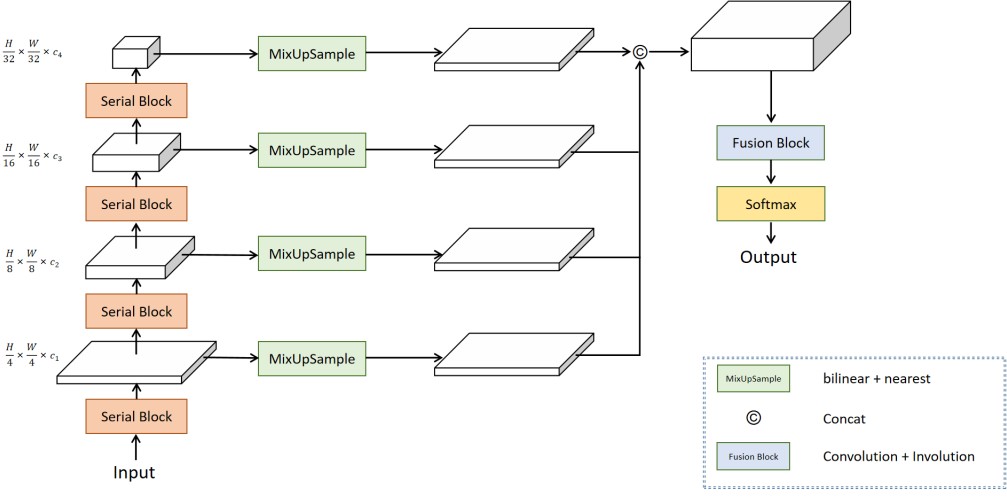

Figure 2: The overall framework of the model proposed in this paper.

The overall structure of CoaT is shown on the left in Fig 2, where the encoder part on the left is CoaT-Lite, through which we can get four different scales of feature maps. the MixUpSample Block samples these different scales of feature maps to get a set of features of the same size and concat in

the channel dimension. We design a Fusion Block to fully fuse these features, and finally, pass a Softmax layer to obtain the final gradient field and binary prediction results.

The structure of the CoaT Serial Block is shown in Fig 3. The module first downsamples the input features through a patch embedding layer, and then flattens them into a set of image tokens sequences. After passing through multiple Conv-Attention modules, the resulting image tokens are reshaped into a 2D feature map input for the next serial block. In addition, the Conv-Attention module is visible on the right side in Fig 3.

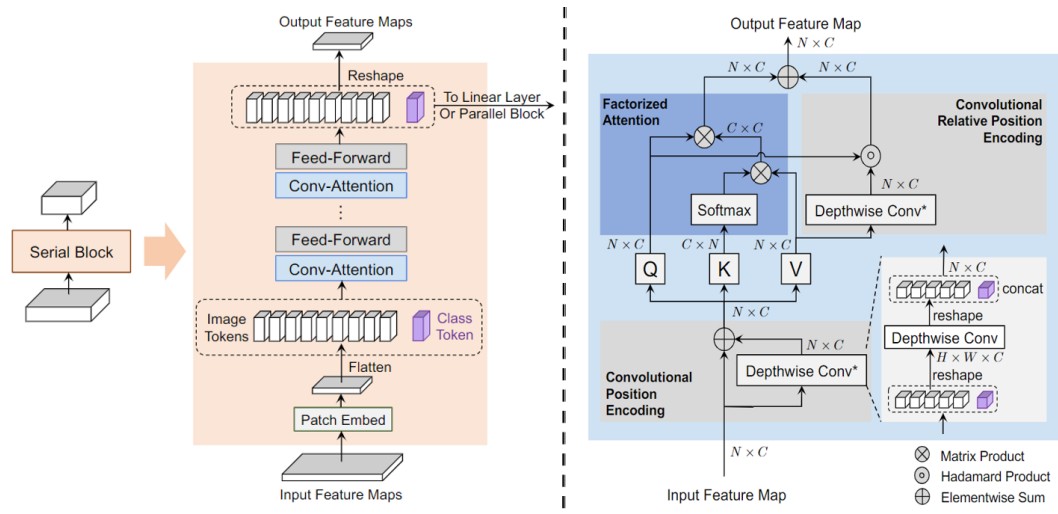

Figure 3: Schematic illustration of the serial block in CoaT and illustration of the conv-attentional module [2].

The MixUpSample Block uses a mixture of bilinear interpolation and nearest neighbor interpolation to upsample feature maps of different sizes to the same size for the subsequent fusion of these feature maps.

Consider that convolution exists spatial-agnostic and channel-specific, while Involution has the opposite property. We use both convolution and involution in Fusion Block so that the feature maps after concat on the channels can be fully integrated in both spatial and channel dimensions.

Loss function: we use the summation between Dice loss and cross entropy loss because compound loss functions have been proved to be robust in various medical image segmentation tasks [3]. In addition, we used MSE loss on the predicted resulting gradient fields.

In order to make full use of the large amount of unlabeled data and to make the model more generalizable, we conducted semi-supervised experiments by the mean teacher [4] approach.

## 2.3   Post-processing

First, the gradient field and binary mask obtained by the backbone network are used to generate a prediction map containing the center pixels's set of each cell instance. It's different from cellpose which marks all pixels inside the cells' masks, because we find that that some center points with similar features are easy to connect into one center point set in the dynamic programming of these center points. To overcome this, we only collect the center region sets using dynamic programming with only 3 iterations (that means only the points within 3 pixels near the center point will participate in the generation of the center point), and then use the Marker-Controlled Watershed Segmentation to finish the instance segmentation.

# 3 Experiments

## 3.1 Dataset

We did not use any real external dataset, but only the official dataset provided by the competition for training and validation, and some rare cell shapes were automatically synthesized offline. The training set consists of 1000 labeled image patches from various microscopic types, tissue types, and stain types, and over 2200 unlabeled images. And there are four microscopy modalities in the training set, including Brightfield, Fluorescent, Phase-contrast, and Differential interference contrast. Noting that some of the cell types in the validation set were fewer in the test set, we used the taichi physics engine [5] to simulate some simply shaped soft cells and generate the cell images automatically. Some of the generated samples are shown in Fig 4. In addition, our encoder is loaded with the weights of the CoaT-Lite Medium version pre-trained under the ImageNet dataset, which can be downloaded at this URL: `https://github.com/mlpc-ucsd/CoaT`.

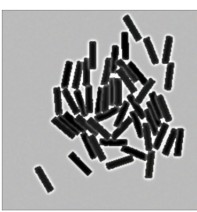 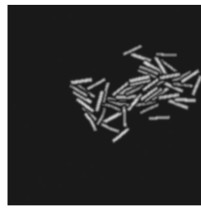 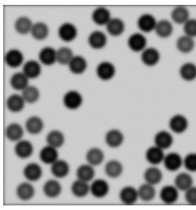 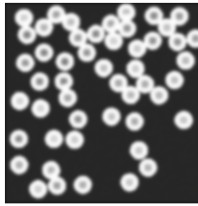

Figure 4: Samples generated by taichi.

## 3.2 Implementation details

Please include all the implementation details in this subsection. The following items are minimal requirements.

### 3.2.1 Environment settings

The development environments and requirements are presented in Table 1.

Table 1: Development environments and requirements.

| System | Ubuntu 18.04.5 LTS |
|---|---|
| CPU | Intel(R) Core(TM) i9-10900X CPU@3.30GHz |
| RAM | 16×4GB; 2.67MT/s |
| GPU (number and type) | 4 NVIDIA 2080Ti 11G |
| CUDA version | 10.2 |
| Programming language | Python 3.8 |
| Deep learning framework | Pytorch (Torch 1.12.0, torchvision 0.13.0) |
| Code | `https://github.com/zzh980123/NeurIPS-CellSeg` |

### 3.2.2 Training protocols

**Data augmentation**
Online:
Official Augmentations: Flip, Rotate, Random Crop, Gauassian Smooth, Gauassian Noise, Adjust Contrast, Histogram Shift, Random Zoom.
More: Color Jitter, Scale Intensity, Cutout.
Offline:
We use the Taichi Physical Engine and mpm18 (Material Point Method) to simulate cell distribution.

patch sampling strategy during training (e.g., randomly sample $512 \times 512$ patches) and inference (slide window with a patch size $768 \times 768$)

optimal model selection criteria

Table 2: Training protocols. If the method includes more than one model, please present this table for each model seperately.

| | |
|---|---|
| Network initialization | "he" normal initialization |
| Patch size | 512×512 |
| Total epochs/Max epochs | 186/1000 |
| Optimizer | AdamW |
| Initial learning rate (lr) | 6e-5 |
| Lr decay schedule | / |
| Training time | <48 hours |
| Loss function | CEDice Loss + MSE Loss |
| Number of model parameters | 50.58M[1] |
| Number of flops | 404.90G[2] |

# 4    Results and discussion

We try to train our model under the framework of mean-teacher and some improved methods of it. But all the method we tried failed (see F1 score below).

By the local validation set and the tuning set, we compare the quantitative of all the cells' segmentation results by F1 score or observation.

We check the validation set output segment results, small cells which area is less than $16 \times N$ (N > 16), and fluorescyte cells always get low F1 score.

Our method follows cellpose, so the total test time is amortized between the model inference stage and the post-processing generation mask stage. The inference time is almost based on the model's FLOPs, so the input size affects much, we tried size of 256, 512, 768 and 1024, the size of 768 get the best score and cost nearly the least time on large images. Besides, a valid way to reduce the inference time is to transfer the model parameters and structure to ONNX/tflite and then use other high-performance inference engines, but no one in our team is good at model deployment (due to the use of a special operator called involution, we failed many times), we give it up. The whole-slide image in our method cost about 1 minute within our vision, therefore we did not try to optimize inference time for such images.

## 4.1    Quantitative results on tuning set

We spend a lot of time exploring the effect of different cell representations and network model structure on segmentation results. Obviously, the SDF-based (Signed Distance Field) cell representation is equivalent to the gradient field based cell representation in theory. In another word, the SDF-based method is is the cumulative function of gradient field based cell in the spatial dimension. But this may case a lower fault tolerance rate for the probability distribution of the center points of the cell with SDF-based method. We compare the different representations at Table 3. The SDF-based model details is in the Appendix.

Some of the experiments on the tuning set are shown in Table 3. Where 3class means the same way as the official baseline provided to get the output of 3 class from the network. From the table, it can be seen that using both gradient field and watershed give the best F1 score in the tuning set. * present without fine-tune at the train dataset. DFC present DeFormable sparse Convolution and the structure is shown at Appendix.

In addition, we performed partial ablation experiments on the fusion module, including using only 3 × 3 convolution, using only Involution, replacing 3 × 3 convolution with an oversized convolution kernel of 31 × 31 size [6, 7]. The results are visible in Table 4. In particular, the experiment with the 31 × 31 convolution kernel size, we used large convolution kernels in multiple places of the decoder instead of just in the fusion block.

Table 3: Quantitative results on tuning set.

| Method | F1 Score |
|---|---|
| MaskRCNN | 74.72 |
| UNet+3class | 54.72 |
| SwinUNetr+3class | 54.72 |
| SwinUNetr+3class+DFC | 56.62 |
| CoaT+3class | 72.30 |
| Cellpose* | 51.42 |
| CoaT+SDF | 58.32 |
| CoaT+grad | 75.45 |
| CoaT+grad+Watershed | 77.24 |

Table 4: Quantitative results of different fusion modules on tuning set.

| Fusion Method | F1 Score |
|---|---|
| $3 \times 3$ Conv | 74.2 |
| $31 \times 31$ Conv [6] | 68.42 |
| Involution [8] | 77.24 |

We also tested the effect of the size of the input feature maps under the model of CoaT+3class, and the results are shown in Table 5. Based on the results of this experiment, we choose 768x768 resolution as the input feature map size for all other models.

Table 5: Quantitative results of different fusion modules on tuning set.

| Input Size | F1 Score |
|---|---|
| $256 \times 256$ | 67.86 |
| $512 \times 512$ | 69.72 |
| $768 \times 768$ | 72.30 |
| $1024 \times 1024$ | 71.90 |

We performed semi-supervised training using the unlabeled dataset using the methods [4, 9] under the mean teacher framework. Perhaps because the distribution of combination of unlabeled images and labeled images is shifted compared with the train/validation dataset's distribution, the results on the tuning set are not satisfactory. Some of the results are shown in Table 6.

Table 6: Quantitative results of different fusion modules on tuning set.

| Semi-supervisory strategy | F1 Score |
|---|---|
| [1]: w/o mean teacher | 77.24 |
| [2]: mean teacher fine tune based [1] | 77.06 |
| [3]: mean teacher + confidence loss | 74.16 |
| [4]: mean teacher + confidence loss + cutout augmentation | 74.17 |

## 4.2 Qualitative results on validation set

Fig 5 presents some easy and hard examples on validation set. Where the first row is the original graph, the second row is the predicted result after full-supervised training, and the third row is the predicted result after semi-supervised training. As you can see from the figure, the semi-supervised approach does not work well.

## 4.3 Segmentation efficiency results on validation set

The average running time is 2.76 s per case in inference phase (totally 127 images in our augment data set), and average used GPU memory is 799.52 MB. The area under GPU memory-time curve is 247845 and the under CPU utilization-time curve is about 1395.

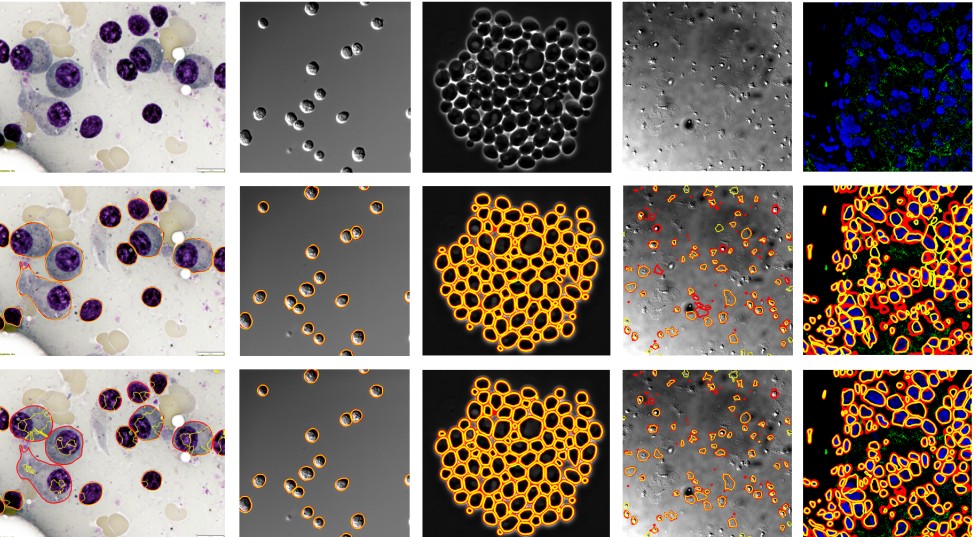

Figure 5: Qualitative results of full-supervised and semi-supervised model on easy (Three columns on the left) and hard examples. Each prediction graph is the result of ground truth and predicted label overlap, where the red outline is the ground truth and the yellow outline is the predicted label obtained.

## 4.4 Limitation and future work

The results of the semi-supervised experiments we have conducted so far are not satisfactory, so we will follow up by learning more about semi-supervision and conducting experiments. We note that omnipose [10] is a proposed method based on cellpose and was recently published in Nature Methods, and we may subsequently refer to these changes to improve the performance of our model. In addition, integrating multiple models and using ONNX to accelerate inference might be a good way to improve the results.

## 5 Conclusion

The main challenge of this task is to separate the adherent cells from each other. Compared to the methods in baseline and cellpose, the method proposed in this paper is significantly improved and better able to accomplish this task.

## Acknowledgement

The authors of this paper declare that the segmentation method they implemented for participation in the NeurIPS 2022 Cell Segmentation challenge has not used any private datasets other than those provided by the organizers and the official external datasets and pretrained models. The proposed solution is fully automatic without any manual intervention.

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

# A  Appendix

**SDF-based representation model**

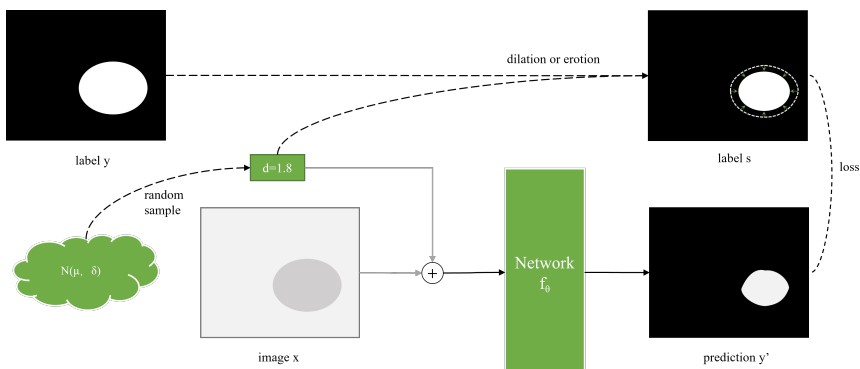

Figure 6: The implicit SDF representation of cell by level set. $x$ and $y$ present the image and label, $d$ and $x$ will be reshaped and concatenated simply before fed into the network.

We experimented with different SDF representations of cell segmentation: explicit SDF and implicit SDF. An implicit sdf is defined as embedding the level set parameter into the network to represent the discrete SDF of the segmentation of cells. Fig 6 shows training protocol of implicit SDF method .

Let $d$ denotes the distance between cell edge and the level, in our experiments, we sampled the $d \in \Omega \in [m, n]$ where $\Omega$ is a truncated normal distribution $\mathcal{N}(\mu, \sigma^2)$, $m$ and $n$ is the boundary of the truncation.

The network learn the function to represent the level set of the segmentation boundary as the Eq.1.

$$f_\theta(x, d) = y' \tag{1}$$

Where, $x$ denotes the image, $y'$ denotes the predict binary classification results match the level $d$, $\theta$ presents the parameters of network.

**Deformable sparse Convolution**

DFC(Deformable sparse Convolution) is similar to DCN(Deformable ConvNets)[11, 12], but the same offset vector is applied for convolution, because the morphology of cells in an image is always similar. The difference is shown in Fig 7.

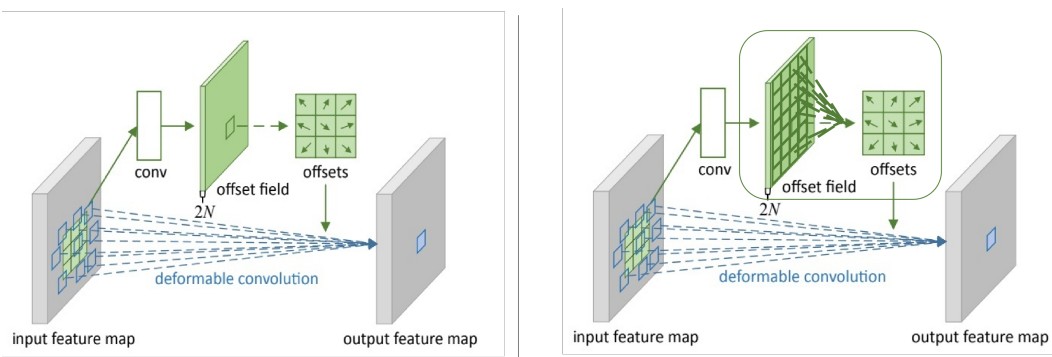

Figure 7: Illustration of the difference between DFC and DCN, every position in the feature map in DFC shares a same offset vector.

