# OpenReview forum: "Watershed-enhanced cell instance segmentation based on guidance of gradient"
_NeurIPS.cc/2022/Challenge/CellSeg — Submitted to NeurIPS CellSeg 2022_

### Official Review · Reviewer_4Hn5 · 2022-12-27
**It's a valid method, but the quality of the writing is poor**

**Rating:** 2
**Confidence:** 4

**Review:**

## **Summary**

The authors present a method for cell segmentation using an encoder-decoder architecture with CoaT[1] as the encoder and a feature fusion block as the decoder. The method is combined with watershed segmentation for pre and post processing and achieves good performance without the use of an external dataset. However, the writing and formatting of the paper are not up to the standards of a methodology paper. In particular, the hand-drawn figure and the inclusion of figures from other papers without proper citation are problematic. The description of the method is insufficient, with key techniques such as watershed segmentation not being adequately explained. The paper is also too short, not meeting the required length of 8-12 pages.

## **Pros**

1. The authors achieve high performance even without external dataset

## **Cons**

1. The writing of the thesis is not only very poor, but the format is not academic. In particular, Figure 1 is a hand-drawn illustration and is very inappropriate.
2. Figures 3 and 7 were taken from the paper [1, 2] as they are.
3. The authors did not follow the rules (8-12 pages) regarding the length of a methodology paper, with a text of 7 pages.
4. The description of the method is too insufficient. For example, there is no description of watershed segmentation, one of the main techniques the authors used.

### References

[1] Xu, Weijian, et al. "Co-scale conv-attentional image transformers." *Proceedings of the IEEE/CVF International Conference on Computer Vision*. 2021.

[2] Dai, Jifeng, et al. "Deformable convolutional networks." *Proceedings of the IEEE international conference on computer vision*. 2017.

---

### Official Review · Reviewer_WL2U · 2023-01-07
**Try to find a good combination of existing methods but the writing is unsatisfied**

**Rating:** 4
**Confidence:** 5

**Review:**

Summary:
The authors present a cell instance segmentation method which achieves F1 score of 0.7724 on the tuning set of CellSeg competition. The authors modified Cellpose’s strategy and constructed a network structure with CoaT as the encoder and a decoder to fuse features with different scales. The segmentation performance was enhanced by using watershed algorithm in post-processing.

Pros:
1.The paper presents four tables to show the effectiveness of different network architectures, cell representations, fusion methods, input sizes and mean teacher. Reporting these results is good.

Cons:
1.The title emphasizes Watershed and the gradient guidance but the details of these two parts are not enough. For example, in Sec. 2.1, how to do ‘Diffusion’? Even though the gradient field method is from another work, it is suggested to present the details to make the paper self-contained.
2.In Sec. 2.3, how to do the dynamic programming? The process of ‘collecting the center region sets’ looks like an erosion operator. The authors describe it as ‘dynamic programming’, but what is the details of the dynamic programming?
3.Why compare with SDF? The motivation is missing. The reference/citation of SDF is also missing.
4.In Tab. 3, why not finetune Cellpose? That makes the comparison unfair and cannot well verify the effectiveness of CoaT. If submitting to the tuning set is limited, it is suggested to split off a validation set for the comparison.
5.For ‘CoaT+grad’ in Tab. 3, when without using the watershed algorithm, how to generate each cell instance mask from the gradient field and the binary mask? In Sec. 2.3, I cannot find these details.
6.The caption of Tab. 6 has typos, ‘different fusion modules. Tab. 6 is about the semi-supervised learning.
7.In Tab. 6, how to compute the confidence loss? The description is missing.
8.In Sec. 1, the authors discuss four difficulties. I cannot understand why the second one is a difficulty: ‘(2) The target cells sizes are relatively balanced over the whole dataset, but the cell size is stable in the same image during cell segmentation.’ As I see, balanced and stable cell sizes usually make the training of neural network easier.
9.In Sec. 1, the authors identify four difficulties. However, they use which techniques to solve which difficulties? I cannot see these descriptions. In my opinion, if we identify and discuss some critical difficulties in the paper, then we should present new or adapt existing techniques to solve these difficulties.

---

### Decision · Program_Chairs · 2023-01-19

Reject